# Identifying publications in questionable journals in the context of performance-based research funding

Joshua Eykens[1]*, Raf Guns[1], A. I. M. Jakaria Rahman[1,2], Tim C. E. Engels[1]

1 Centre for R&D Monitoring (ECOOM), Faculty of Social Sciences, University of Antwerp, Antwerp, Belgium,
2 Department of Communication and Learning in Science, Chalmers University of Technology, Gothenburg, Sweden

* joshua.eykens@uantwerpen.be

**Data Availability Statement:** The data can be found at: https://github.com/JoshuaE1/Questionable-journals-in-PRFS/tree/master/data; https://zenodo.org/record/2842559#. XOexmogzZwM.

## Abstract

In this article we discuss the five yearly screenings for publications in questionable journals which have been carried out in the context of the performance-based research funding model in Flanders, Belgium. The Flemish funding model expanded from 2010 onwards, with a comprehensive bibliographic database for research output in the social sciences and humanities. Along with an overview of the procedures followed during the screenings for articles in questionable journals submitted for inclusion in this database, we present a bibliographic analysis of the publications identified. First, we show how the yearly number of publications in questionable journals has evolved over the period 2003–2016. Second, we present a disciplinary classification of the identified journals. In the third part of the results section, three authorship characteristics are discussed: multi-authorship, the seniority–or experience level–of authors in general and of the first author in particular, and the relation of the disciplinary scope of the journal (cognitive classification) with the departmental affiliation of the authors (organizational classification). Our results regarding yearly rates of publications in questionable journals indicate that awareness of the risks of questionable journals does not lead to a turn away from open access in general. The number of publications in open access journals rises every year, while the number of publications in questionable journals decreases from 2012 onwards. We find further that both early career and more senior researchers publish in questionable journals. We show that the average proportion of senior authors contributing to publications in questionable journals is somewhat higher than that for publications in open access journals. In addition, this paper yields insight into the extent to which publications in questionable journals pose a threat to the public and political legitimacy of a performance-based research funding system of a western European region. We include concrete suggestions for those tasked with maintaining bibliographic databases and screening for publications in questionable journals.

**Funding:** This investigation has been made possible by the financial support of the Flemish government to the Centre for R&D Monitoring (ECOOM). The funders had no role in study design, data collection and analysis, decision to publish, or preparation of the manuscript.

**Competing interests:** The authors have declared that no competing interests exist.

## 1. Introduction

Predatory open access (POA) publishing has recently gained a lot of attention in mainstream media across the world [1, 2]. A report released by the International Consortium of Investigative Journalists (ICIJ) publicized the results of an analysis of 175,000 scientific papers which were published by two of the world's largest pseudo-scientific platforms [3]. The consortium observed that researchers from all over the world, including authors affiliated with universities in Flanders, had published in questionable journals. For almost a decade, the main issues related to publishing questionable journals have been known to academics and research policy makers [4].

The evaluation of (new) knowledge is one of the crucial and systemic components that differentiates science from other domains in society [5]. Peer review of scientific manuscripts, one of the corner stones of scientific communication, as such is deemed to be crucial. Therefore, in many contexts, peer review of scientific publications is a standard for research output to be considered as 'scientific'. Research policy makers are increasingly integrating this perception in their regulations as well, e.g. in many performance-based research funding systems (PRFS) [6]. This is also the case for Flanders; in order for publications to be counted in the Flemish PRFS, the criterion of verifiable peer review needs to be met [7].

Questionable open access publishing can be a threat to the legitimacy of PRFSs, as the journals and publishers involved do not abide by best practices in academic publishing (i.e. they purport to carry out peer review and related quality assurance but by and large fail to do so in practice) [8]. Taking into account questionable journals and publishers in PRFSs can do serious harm to the legitimacy and trustworthiness of the funding distribution as well as to evaluations that use the outcomes of PRFSs as a basis [9]. This is probably especially true when a PRFS is wholly or partly based on a national or regional bibliographic database [10, 11] since these do not benefit from an aura of 'excellence' as do the global commercial citation databases like Web of Science (WoS) and Scopus.

In this article we discuss and analyze the particular case of screening for publications in questionable journals submitted for inclusion in a regional bibliographic database for the social sciences and humanities (SSH), namely the VABB-SHW. The VABB-SHW is a comprehensive bibliographic database covering the metadata of publications authored by researchers affiliated to a university SSH unit in Flanders, Belgium. It is a component of the university funding allocation model of the Flemish government and has been introduced in order to better account for SSH research in the Flemish PRFS [12]. Because of the problems surrounding questionable open access journals, ECOOM, the Flemish Centre for R&D Monitoring which is responsible for the maintenance of the VABB-SHW, scans the database for publications in journals considered as being POA on a yearly basis. For clarity, in the remainder of this text, we will refer to questionable or fraudulent journals and publications therein as POA journals or POA publications, but note that the terminology (e.g. 'predatory' open access) is contested [13].

## 2. Background

After a short introduction to the workings of the VABB-SHW database in section 2.1, section 2.2 provides an overview of the five screenings that were conducted and the procedures followed. We point out some difficulties that were encountered and a key lesson learned regarding black- and whitelists. The paper proceeds with a short literature review (section 2.3) which motivated our analysis of the characteristics of the flagged papers and their authors. More specifically, we look into the yearly number of publications that appeared in POA journals. We contrast these with the yearly number of peer-reviewed publications and the number of

publication in journals indexed by the Directory of Open Access Journals (DOAJ). We explore questions regarding authorship characteristics (e.g. number of multi-authored papers, experience level of authors, departmental affiliation) as well as disciplinary classifications of the journals. The present study is an extended version of conference proceeding presented at the Science and Technology Indicators conference, Leiden (The Netherlands), September 2018 [14]. In this paper, we elaborate further on the short, aggregate overview given in the conference proceeding and present novel bibliographic analyses of the publications identified.

## 2.1. VABB-SHW, The Flemish bibliographic database

The VABB-SHW is a comprehensive bibliographic database that contains all publications belonging to five publication types (journal articles, monographs, edited books, book chapters, and proceedings papers), and authored by researchers affiliated to an SSH unit at a Flemish university. For a full description of the VABB-SHW in the context of the PRFS see [7, 12]. It is important to note that scholars sometimes hold multiple affiliations. This leads to the fact that the output covered in VABB-SHW is not exclusively related to the cognitive areas pertaining to the SSH. We will come back to this further on. Additionally, details on discrepancies between organizational and cognitive classifications of publications in VABB-SHW can be consulted in [15].

Every year the Flemish universities submit the metadata of new publications from the past two years to ECOOM for potential inclusion in the VABB-SHW. A distinction is made between publications that are indexed in WoS and those that are not, since WoS-indexed publications are counted in a separate parameter of the Flemish PRFS [7]. Publications that are not indexed in WoS can be counted in the VABB-SHW parameter of the PRFS if they:

1. *are publicly accessible;*

2. *are unambiguously identifiable by an ISSN and/or ISBN;*

3. *contribute to the development of new insights or the application thereof;*

4. *have been peer-reviewed by independent experts in the field prior to publication.*

5. *count at least four pages.*

A panel of independent scholars oversees the selection of publications for the VABB-SHW ('Gezaghebbende Panel' or GP). More specifically, the GP annually decides the publications with which journals and publishers are to be included in the VABB-SHW. Given doubts about the extent to which some questionable journals meet the criteria 3 and 4 the GP requested to cross-check the annual journal list with available black- and whitelists to screen for potential POA journals from 2014 onwards [16].

## 2.2. Screening VABB-SHW for publications in questionable journals

Each of the five consecutive annual screening reports is publicly available [16–20]. Table 1 shows in the third and fourth columns the numbers of journals and articles that were identified as potentially predatory during the screenings. These counts should be interpreted with care, as they represent the number of journals (and VABB-SHW indexed publications in them) that were found on the blacklist, without taking the occurrence on whitelists into account.

The first two screening rounds of the submissions for the VABB-SHW (i.e. editions IV and V) only compared the VABB-SHW journal list to Beall's lists and WoS. At that time, the lists of predatory publishers and journals maintained by Jeffrey Beall, commonly known as "Beall's

**Table 1. Numbers of journals and articles identified for each version of the VABB-SHW, and sources used to identify them.**

| Publication time span | VABB-SHW edition | Journals on blacklist (N) | Publications in blacklisted journals (N) | Blacklist used | Whitelist used |
|---|---|---|---|---|---|
| **2003–2012** | IV | 62 | 59[a] | Beall's lists | WoS |
| **2004–2013** | V | 109 | 138 | Beall's lists | WoS |
| **2005–2014** | VI | 128 | 315 | Beall's lists | DOAJ & WoS |
| **2006–2015** | VII | 185 | 501 | Beall's lists | DOAJ & WoS |
| **2007–2016** | VIII | 65 | 91 | Cabell's Journal Blacklist | DOAJ & WoS |

[a] Note that the number of articles is lower than the number of journals. This results from comparing the raw VABB-SHW journal list with Beall's lists. We found three journals on our journal list which were also present on Beall's list, but no publication appeared in these journals for the predefined time window. For details see [16].

lists", were arguably the most widely used references to identify predatory publishers [21]. In 2016, the Australian Business Deans Council, for example, removed journals from their Journal Quality List based on the information available on Beall's lists [22].

From Beall's lists, publisher names and individual journal titles were collected and matched with the data submitted for potential inclusion in the VABB-SHW. All matches were manually double-checked by the staff of ECOOM [16–19]. Based on the information available through the VABB-SHW, it was also checked which journals were indexed in WoS and which journals had previously been classified as either peer-reviewed or not peer-reviewed by the GP.

After the first two reports, it became clear that there is a great deal of ambiguity in a screening exercise of this kind. It seems reasonable to assume that if a journal or a publishing house appears on Beall's lists (and hence, is potentially predatory), such communication channels do not 'make it through' WoS' indexing services, which is commonly credited for having high standards in journal selection procedures. The selection process for indexation in WoS revolves around four central factors (https://clarivate.com/essays/journal-selection-process/);

1. *Basic publishing standards*: Peer review practices, existence of acknowledgments sections, (un)ethical publishing practices (such as inauthentic journal self-citations), publishing format, etc. The editorial board must be following international editorial conventions (i.e. informative journal titles, full descriptive article titles, author abstracts, complete bibliographic information, and full address information for every author;

2. *Editorial content*: The editorial staff of Clarivate determines if a journal will enrich the database or if the topic is already adequately addressed in existing coverage;

3. *International focus*: Clarivate's editorial staff also assesses the diversity among the journal's contributing authors, editors, and editorial advisory board. For regionally oriented journals it is expected that they contain English-language bibliographic information;

4. *Citation analysis*: Citation analysis is used to determine the importance and influence of a journal in the surrounding literature of its subject.

However, submissions for VABB-SHW versions IV and V included 16 and 17 journals respectively which were indexed both by WoS and Beall's lists (version V included the same 16 journals as version IV plus one extra). Although low in numbers, they do highlight the ambiguity and criticism surrounding the identification of questionable journals with the use of black- and whitelists.

Jeffrey Beall himself linked the emergence of POA to the rise of the Gold OA publishing model [23]. He stated that the transition to an OA publication model would introduce new challenges as well as promising new opportunities [24]. Although he did a remarkable job in compiling information on POA journals and publishers, this did not happen without criticism

from many angles, which in some cases forced him to re-evaluate decisions made, and even remove publishers from his list [25].

For our purposes, these factors highlighted the need to look for an extra source list. To cross-validate the blacklisting by Jeffrey Beall, and thus account for the difficulties related to it, the DOAJ (https://doaj.org) was consulted as an additional source for the next three screenings [18–20]. The DOAJ, like WoS, makes use of a set of criteria (more than 40) to assess the quality of journals listed. In a recent post by the editor in chief of the DOAJ the evaluation procedure is further detailed [26].

The addition of this 'whitelisting' method exemplifies concerns regarding the use of black- and whitelists, e.g. different lists use different criteria and thus should be used as indicative sources [27]. It showed that there is both an overlap between Beall's lists and DOAJ, and also between Beall's lists and WoS. Submissions for VABB-SHW version VI contained 13 journals that were on Beall's lists, despite being indexed by both WoS and the DOAJ. A closer look made clear that these were all journals of the much-debated Swiss publishing house Frontiers [25, 28].

In the course of January 2017, Beall's list and website were taken offline, and from that moment onwards, Beall's lists are no longer maintained [28]. Thus, for the screening of version VIII, Cabells' Journal Blacklist (henceforth CJB, https://www2.cabells.com/blacklist) is used, together with the DOAJ as a whitelist. CJB is a commercial service provided by Cabells Scholarly Analytics. It has a few practical advantages compared to Beall's lists. First, it is a list of journals rather than publishers. Second, it allows for lookup by ISSN, which is more reliable than title-based comparisons. Third, CJB also tracks questionable journals other than those that provide open-access policies or formats (i.e. closed-access). Upon inspection such cases, however, are not present in our data.

Third, CJB provides reports listing violations when information on a specific journal is consulted. Jeffrey Beall also made use of a list of criteria to identify predatory publishing houses, but Cabells Scholarly Analytics does this in a structured, pre-determined and, more importantly, transparent fashion[2]. It is remarkable how the total number of questionable journals that were identified during the previous screenings dropped by 120 between versions VII and VIII (see Table 1). The main lesson learned from these screenings relates to the dynamicity of the journal landscape. The usage of black- and whitelists alike should be done with considerable care. Final validation by experts in this regard, is crucial. We now turn to a short literature review which motivated our analyses of the articles that were identified during the screenings and rejected based on their questionable nature by the GP.

## 2.3. Literature review

According to Xia et al. (2015) and Frandsen (2017) authors who publish in POA journals as well as researchers citing such articles are typically "inexperienced authors from Africa, Southeast Asia or South Asia and to a lesser extent experienced authors from the rest of the world" [29, 30]. Shen and Björk [31], however, report that their sample of authors contains a sizable subset of researchers from Europe (8.8%), North America (9.2%), and Australia (1.5%). In another sample of 1,907 POA papers, more than half were found to have authors from upper-middle-income to higher-come countries [32].

The latter studies suggest that, while the majority of authors who published in POA journals are geographically concentrated in developing countries, also authors from other countries are involved. Most research has focused on sampling a set of journals from e.g. Beall's lists, but comparatively little is known about the extent of POA publishing by authors working in West-European countries. It has been suggested that the problem of POA publishing is relatively

small and that overstating the scope may lead to discrediting open access as a whole [33]. This raises the question whether initiatives to raise awareness about POA publishing have the adverse effect of tarnishing all open access journals with the same brush.

On the basis of a survey sent to corresponding authors of 300 papers in POA journals, Kurt [34] has identified four themes as major drivers behind publishing in POA journals: social identity threat (fear of being regarded as inferior when submitting to a Western journal), lack of research proficiency, unawareness of their POA nature, and pressure to publish. While Kurt [34] explicitly links the first two themes to authors from developing countries, the last two may also manifest in Western countries.

Unawareness has been linked to lack of experience [34], but can senior authors distinguish more easily between legitimate and questionable journals? Some factors that might contribute to inadvertently publishing in a POA journal are collaboration–relying on the judgement of one's co-authors–and multidisciplinarity–publishing in a field outside of one's core field. Indeed, based on the observation that senior authors are more prolific authors in general, one could argue that it is to be expected that they also contribute more to POA publications. Wallace and Perri [35] show for a sample of papers in the field of economics, that also more experienced scholars do publish in POA journals. Younger researchers on their turn, seem to be more positive towards open access outlets [36], and may thus be more likely to publish their findings in them. Extrapolating from this, one might expect younger researchers to be more likely to send their work to questionable open access journals. In balance, both junior and senior researchers might be equally implied in POA publishing.

Pressure to publish is commonly associated with new public management in academia and performance-based research funding in particular [37]. Indeed, confronted with implicit and/or explicit expectations of publishing productivity some researchers might turn to POA publishing as a tactic to increase their output [35]. As such publishing in POA journals may go hand in glove with other often raised issues such as salami-slicing, risk averseness, and undue self-citations.

Based on our experience with five consecutive annual screenings of the VABB-SHW, a bibliographic database that feeds into the Flemish PRFS and provides full coverage of the region's output in the SSH, this paper addresses the following research questions:

**Q1.** What is the yearly number and evolution of POA journals and publications in Flemish SSH, and how does this compare to the number and evolution of legitimate gold OA journals and publications as well as to the total number of peer reviewed publications in the PRFS?

**Q2.** How are POA publications distributed over fields?

**Q3.** What are the authorship characteristics of POA publications?

**Q3.1** Are these publications single- or multi-authored, and how does this relate to the set of gold OA publications and to the entire dataset?

**Q3.2** Are these publications written by senior or junior researchers?

**Q3.3** Are junior researchers more likely to occupy the first position in case of multi-authored POA publications?

**Q3.4** Are the authors working in the same field as the journal is situated in?

The results of this analysis are unique in that the issue of POA is studied on the basis of a regional full coverage dataset that is part of the regional PRFS. We now discuss and show how we addressed each separate research question with the data at hand.

## 3. Methods

Based on ECOOM's five reports, a list of all identified, potential POA journals was made. Subsequently, all publications that appeared in these journals and were submitted for inclusion in an edition of the VABB-SHW were identified (N = 556). As explained above, the 'predatory' nature of several of these journals is contested. Moreover, both the blacklists and whitelists used, especially Beall's list and DOAJ, have seen various updates: journals and publishers have been added or removed at various time points. Together, these issues called for a clearer operationalization of the 'truly problematic' cases. To tackle this, we have used the decisions by the GP (for each individual journal) as a benchmark. Note that this list of journals which were selected by the GP is not the same as those that were used for our screenings. The black- and white lists used for a screening round are at the time of the screening kept as provided by the mentioned web services (e.g. Web of Science, Cabells, Beall's lists, DOAJ). If a journal on the VABB-SHW journal list is listed by a blacklisting service during a screening round, it is presented as such in the reports. Based on these reports, final decisions are made by the GP as to whether a journal is really problematic. If so, the journal is listed as non-peer reviewed and labeled as POA. When after a number of years another article appears in the same journal and that journal is not listed by any blacklisting service, the GP re-evaluates their decision. In the case of journals appearing on both black- and white lists, it is the GP that decides whether the journal should be regarded as peer reviewed or not. The list used for the analyses presented here is the result of the final selection by the GP as of 2018. All data and code for the analysis are openly available [38].

**Q1:** What is the yearly number and evolution of POA journals and publications in Flemish SSH, and how does this compare to the number and evolution of legitimate gold OA journals and publications as well as to the total number of peer reviewed publications in the PRFS?

The VABB-SHW database (publication years 2003–2016) has been checked for publications that are flagged as POA by the GP. In total, this procedure yields 210 rejected publications. They have appeared in 144 unique journals. The number of legitimate OA journals (and publications therein) present in VABB-SHW is approximated by checking each journal's indexation in the DOAJ (data from March 22, 2018). Since the DOAJ upholds a set of basic requirements for inclusion, the journals on this list can be considered to adhere to the most important good practices in (OA) academic publishing. It should be stressed that this approach does not exhaustively cover all OA publications, the total number of which is probably far larger; most importantly, it does not cover self-archived preprints and post prints (Green OA). The set of POA journals and articles thus correspond to the journals which were selected and marked by the GP as being POA. These journals (publications) are not considered 'peer-reviewed' and as such are not a part of the set of peer-reviewed or 'in-DOAJ' journals and articles, regardless of the fact that they might appear on the list of DOAJ indexed journals. The list of DOAJ-indexed journals was compiled by comparing the ISSNs of peer-reviewed journals in our database with DOAJ's journal list.

**Q2:** How are POA publications distributed over fields?

In the VABB-SHW, publications are classified according to the institutional affiliation of the authors. That is, the VABB-SHW uses an organizational disciplinary classification [15]. This implies that a publication can have multiple disciplines assigned, because of co-authorship and multiple affiliations. Moreover, a sizable number of articles (co-)authored by researchers affiliated to SSH units can be classified in non-SSH field like medicine or natural sciences. To spot field-related differences, all journals were classified according to the Fields of

Science classification of the Organisation for Economic Co-operation and Development [15, 39], following the procedures explained in [15].

**Q3.** What are the authorship characteristics of POA publications?

The list of 210 POA publications yields a list of 808 authors, corresponding to an average of almost four authors per article (3.85). These include 341 unique authors that are affiliated to a Flemish university at the time of writing. The analysis presented here focuses on these 341 authors since we have additional, disambiguated information concerning their publication records, including their departmental affiliation. We do not have detailed information on their employment status or current career stage.

As stated in section 2, earlier research found important differences when it comes to differences in publication behavior of junior vs. senior scholars. To give us some understanding of the authorship characteristics of the rejected publications, we will address four different questions relating to single- and multi-authorship (3.1), the difference between junior and senior authors with regard to position in the author-byline (3.2 and 3.3), and overlapping of disciplinary scope of the authors and the journals they publish in (3.4).

**Q3.1:** Are these publications single- or multi-authored, and how does this relate to the set of gold OA publications and to the entire dataset?

For each publication in the database, we count the total number of authors in the author list. The records of POA publications have been manually compared to online information on these publications, to ensure their accuracy. Some articles, however, are not digitally accessible anymore.

**Q3.2:** Are these publications written/co-authored by senior or junior researchers, and how does this relate to the entire dataset?

We have divided the authors present in our database into two groups. The authors who are considered as seniors are those that have published at least ten publications and in at least five different publication years. All others are considered junior. This operationalization of seniority is based on the one that is currently used by the Research Foundation Flanders (FWO) and is similar to the definition used in earlier studies based on the VABB-SHW [40]. It enables us to distinguish between early stage researchers (i.e. pre-doctoral or early stage postdoctoral level researchers) and more senior staff. At the Flemish universities, doctoral researchers, for example, are usually not appointed for more than four years. Having published over a period of five or more years thus assumes a prolonged career in academia. It follows that the authors classified as 'senior' are expected to have reasonable experience when it comes to journal selection and related publication procedures.

Based on this information, for each article, we calculate the proportion of identified seniors relative to all identified authors in the author byline of the article. Note that we only consider identified authors and disregard all non-Flemish co-authors, since their publications are not systematically tracked. For each publication year, we calculate the average proportion for all publications combined. Formally, the average proportion *P* of senior authors who have contributed to a set of publications for a given publication year (in our case: POA, DOAJ-indexed or peer-reviewed) is determined as follows:

$$P = \frac{1}{N} \sum_{i=1}^{N} \sum_{j=1}^{|A_i|} \frac{a_j^i}{|A_i|}$$

Here, N is the number of publications, $A_i$ is the set of identified authors $\{a_1^i, \dots, a_{|A_i|}^i\}$ of publication $i$, and $|A_i|$ is the number of identified authors of publication $i$. $a_j^i = 1$ if author $j$ is a senior author, and 0 otherwise. $P$ ranges from 0 (no contribution by senior authors) to 1 (no contribution by junior authors).

**Q3.3:** Are junior researchers more likely to occupy the first position in case of multi-authored POA publications?

We look at the proportion of cases in which a junior or senior author is positioned first, compared to taking a position in the remainder of the byline. In medicine, for example, the first author of a publication is often considered the one who contributed most to a manuscript [41].

**Q3.4:** Are the authors working in the same field as the journal is situated in?

To check for this, we compare the list of journals and their respective disciplinary classification (i.e. cognitive classification) with the organizational classification (i.e. based on the affiliation of the authors) of the articles. This is done manually. For a detailed description on these classification methods, we refer to Guns et al. [15]. We create a matrix, with cognitive disciplines as rows and organizational disciplines as columns. Each cell contains the number of publications that has the given combination of cognitive and organizational discipline.

## 4. Results

### 4.1. Open Access (OA) and POA publications in VABB-SHW

In total, we have identified 210 POA journal articles which could be linked to 144 unique POA journals. Table 2 shows the number of publications and journals per publication year until 2016. The results presented demonstrate a relatively steep increase in POA journals and publications that occurred between 2009 and 2012, which is followed by a decline from 2012 onwards. This decline from 2012–2013 onwards looks promising. This might be the result of a growing awareness among researchers of the problem of POA, due to the awareness raising campaigns by the universities as well as other initiatives like the yearly ECOOM reports.

**Table 2. Counts of POA, DOAJ indexed and peer-reviewed articles and journals.**

| Year | POA | | DOAJ indexed | | Peer-reviewed | |
|---|---|---|---|---|---|---|
| | Publications | Journals | Publications | Journals | Publications | Journals |
| 2003 | 0 | 0 | 72 | 39 | 2924 | 1324 |
| 2004 | 2 | 2 | 75 | 50 | 3305 | 1586 |
| 2005 | 4 | 4 | 109 | 59 | 3559 | 1703 |
| 2006 | 8 | 6 | 119 | 67 | 3870 | 1811 |
| 2007 | 9 | 8 | 162 | 94 | 4040 | 1976 |
| 2008 | 8 | 7 | 195 | 99 | 4506 | 2208 |
| 2009 | 10 | 10 | 260 | 142 | 4751 | 2382 |
| 2010 | 17 | 16 | 273 | 158 | 4818 | 2392 |
| 2011 | 27 | 25 | 407 | 208 | 5220 | 2601 |
| 2012 | 48 | 42 | 471 | 235 | 5542 | 2723 |
| 2013 | 35 | 32 | 589 | 270 | 5784 | 2891 |
| 2014 | 32 | 27 | 690 | 315 | 6048 | 3065 |
| 2015 | 6 | 5 | 808 | 352 | 6289 | 3189 |
| 2016 | 4 | 4 | 784 | 357 | 5327 | 2931 |

A second observation relates to the journals indexed by the DOAJ. For the Gold OA articles and journals, we observe a clear increase for each consecutive publication year. The number of journals indexed by the DOAJ for publication year 2014 is six times as large as the frequency for 2004. The frequency of articles in OA journals in 2014 is almost ten times as large as for 2004. These results suggest that scholars can distinguish between predatory and legitimate OA and, although the numbers of POA publications and journals have dropped after 2012, they are not turning away from OA journals in general. With regard to the importance of individual OA journals, we observe that the average number of articles per journal has increased from 1.8 to 2.3 over the 2003–2015 time window.

## 4.2. Disciplinary classification of POA journals by Fields of Science (OECD)

When looking more closely at the flagged journals and their discipline(s) (Fig 1), some noticeable differences were found. Keeping in mind that we are studying a database for research output mainly from the SSH, it is remarkable that there are actually more journals (59.7%) belonging to the natural, engineering, medical, and agricultural sciences. Journals related to medical sciences account for 33.3% of the total set. This is in line with the findings of earlier studies: Moher and Srivastava [42] have found most 'predatory' publishers are active in the field of biomedical research and Manca et al. [43] show that POA is also a serious problem in the neurosciences and related fields.

Among the journals classified as SSH, we observe that journals related to business and economics are the most prevalent. There is only one humanities journal, which was classified in the field of Philosophy. Some journals could not be assigned to a single field. They were rather broad in scope, and classified as 'general/very broad titles' (9.7%). Examples of these cases are 'Scientific Research and Essays' and 'Nature and Science'. For a more detailed plot, we refer the reader to the supporting information section, S1 Fig.

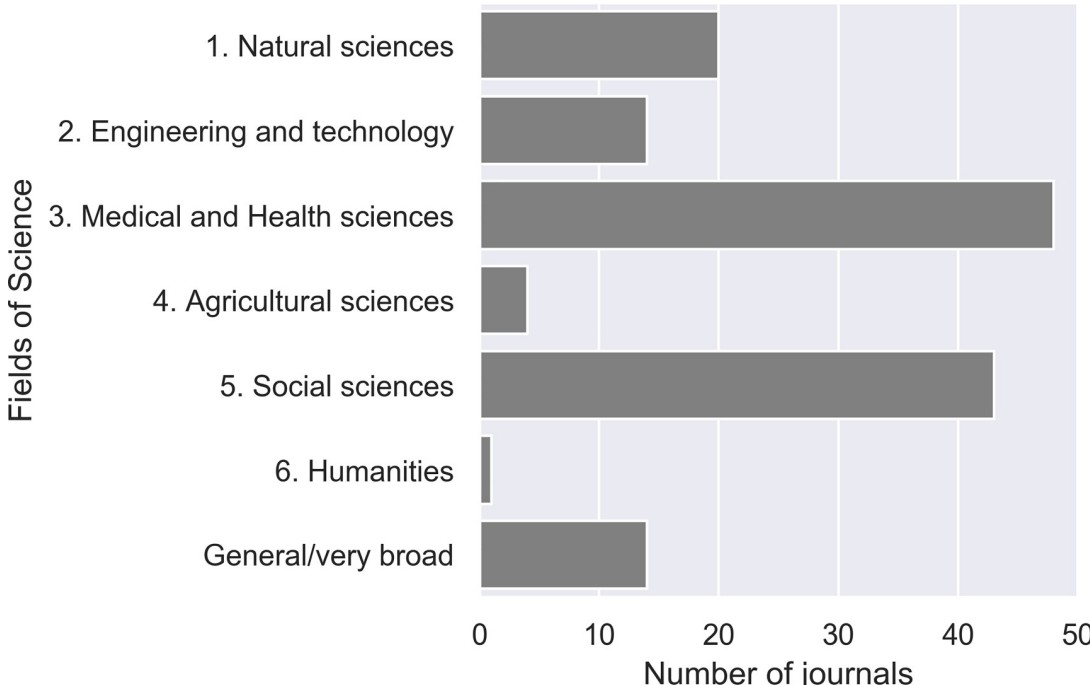

**Fig 1. Distribution of POA journals (absolute counts) classified by Fields of Science coding scheme (N = 144).**

**Table 3. Number (and column-wise percentage) of POA, DOAJ indexed and peer-reviewed publications with N authors.**

| Number of authors listed | POA publications | DOAJ indexed publications | Peer-reviewed publications |
|---|---|---|---|
| 1 | 23 (10.95) | 1141 (22.15) | 27230 (36.95) |
| 2 | 46 (21.90) | 666 (12.93) | 13212 (17.93) |
| 3 | 41 (19.52) | 670 (13.01) | 9607 (13.04) |
| 4 | 39 (18.57) | 606 (11.76) | 6897 (9.36) |
| 5+ | 61 (29.06) | 2068 (40.15) | 16748 (22.72) |
| Total | 210 (100) | 5151 (100) | 73694 (100) |

## 4.3. Authorship characteristics

**4.3.1. Single- and multi-authored publications.** Table 3 shows the counts of articles with 1, 2, 3, 4, and 5 or more authors. As we have shown in the previous section, a relatively large share of the publications appeared in journals which could be considered as non-SSH (e.g. natural sciences, engineering, medical sciences). Given what we already know about authorship patterns in, for example, biomedicine, we would expect to find a lot of multi-authored publications. Indeed, only a small share of the articles (10.48%) are single authored, with the majority of publications having 2 to 14 authors (89.52%). Among the publications with 5 or more authors, there are almost no publications which are classified (cognitive classification) as SSH research: besides two publications in the field social and economic geography and one in the field of economics and business, all others are in non-SSH fields.

We did a $X^2$-test to compare the frequencies of articles with n authors in the entire database. The results indicate that there is a difference (2, N = 79055, 65.85, $p < .001$) between the two distributions. Overall, we can conclude that multi-authored publications are more prevalent for the POA case. The distribution of publications in DOAJ indexed journals and the distribution of POA publications also differ significantly from each other (2, N = 5361, 44.38, $p < .001$). Note that a very large share of publications in DOAJ indexed journals has 5 or more authors, both in comparison with publications in POA journals and in other peer-reviewed journals.

**4.3.2. Contributions by junior or senior staff.** Table 4 lists the average proportion of senior authors per publication for POA publications, publications in DOAJ indexed journals,

**Table 4. Average proportion of senior authors relative to all authors (senior and junior) for POA publications, publications in journals indexed by DOAJ and for peer-reviewed publications.**

| Year | POA publications | DOAJ indexed publications | Peer-reviewed publications |
|---|---|---|---|
| 2003 | – | 0.64 | 0.76 |
| 2004 | 1.00 | 0.77 | 0.81 |
| 2005 | 0.75 | 0.77 | 0.80 |
| 2006 | 0.95 | 0.70 | 0.81 |
| 2007 | 0.76 | 0.75 | 0.81 |
| 2008 | 0.77 | 0.73 | 0.81 |
| 2009 | 0.81 | 0.72 | 0.79 |
| 2010 | 0.91 | 0.71 | 0.80 |
| 2011 | 0.86 | 0.74 | 0.79 |
| 2012 | 0.67 | 0.70 | 0.77 |
| 2013 | 0.60 | 0.66 | 0.73 |
| 2014 | 0.69 | 0.63 | 0.67 |
| 2015 | 0.63 | 0.58 | 0.62 |
| 2016 | 0.75 | 0.56 | 0.60 |

and all other peer-reviewed journals. As one might expect, we observe that in all cases the proportion of senior authors is greater than that of juniors. Contrary to what one might expect, however, we observe that the proportions of senior authors listed on POA publications is in 8 out of 13 of the publication years higher than the number of those listed on other peer-reviewed publications. Whereas the proportion of senior authors on the byline of publications in journals indexed by DOAJ is in all cases lower than that of the peer-reviewed journals. The latter seems to confirm the findings of Nicholas et al. [36]–junior researchers could show a more positive stance towards OA journals in general.

**4.3.3. The position of juniors and seniors in the author list.** The bins in the plot displayed in Fig 2 represent the share of cases in which a junior (light grey) is holding the first position in the byline compared to that share for senior authors (dark grey). Note that the sum of the shares for junior and senior authors is not equal to 100%: the remaining cases involve an (first) author who is not affiliated to a university in Flanders.

In the case of a publication with two authors, junior and senior authors are first author in approximately the same percentage of cases. When comparing these results with the articles

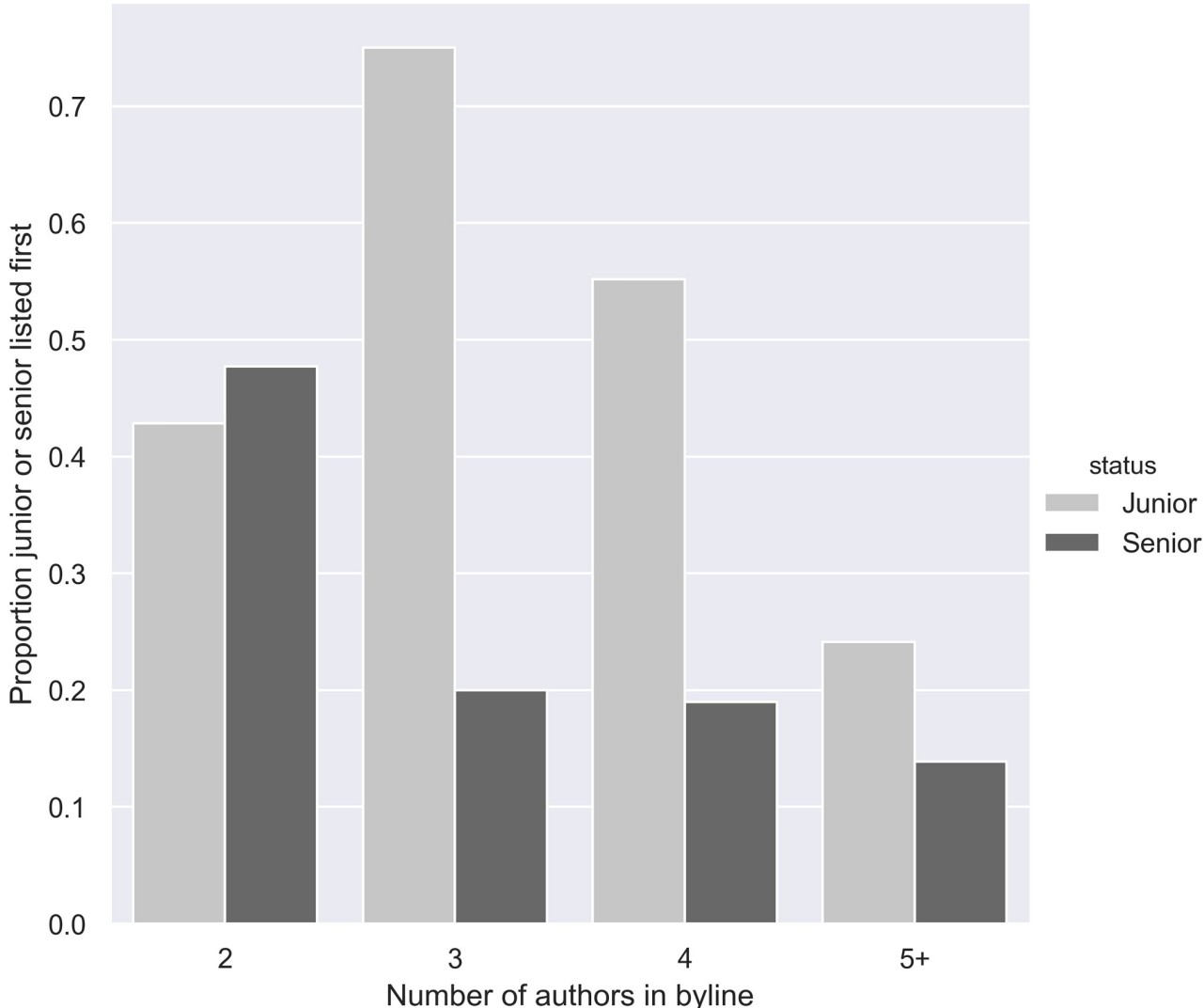

**Fig 2. Proportion of junior vs. senior authors positioned first in the author byline of POA publications.**

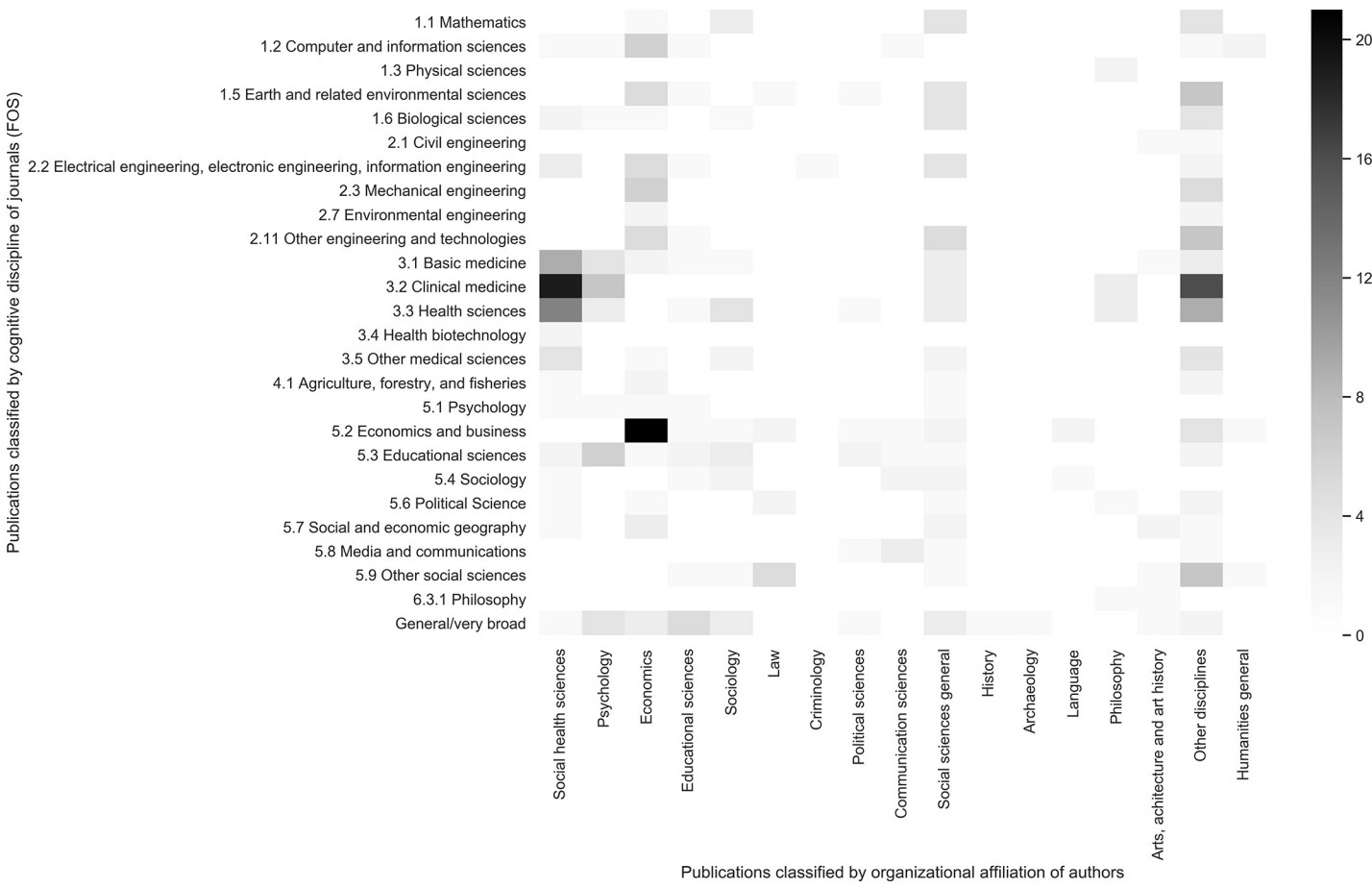

**Fig 3. Heatmap of publications classified by cognitive discipline of journals (rows) by publications classified by organizational affiliation of the authors (columns).**

with three or more authors, we note that the senior authors generally hold second place or other positions further down the byline. In 75% of the cases in which a junior scholar contributed to a publication with 3 authors, he or she was holding the first position in the author list. We wish to emphasize that alphabetically ordered bylines are common within the field of business and economics. Because of the limited size of our sample, however, we could not use the model that estimates the share of intentional alphabetical co-authorship employed in [44].

**4.3.4. Are the authors working in the same field as the journal is situated in?.** Now we turn to the question whether the authors who have published in a POA journal are working in the same field (i.e. organizational classification of publications) as the journal is situated in (i.e. cognitive classification of publications). Overall, there appears to be a rather high overlapping of the two disciplinary indicators in our data if we look at authors affiliated to a (social) health sciences unit and the medical related journals. The same goes for economics and business. The other cases, however, exhibit dispersed overlapping with other fields.

Two exceptions can thus be observed for publications which are organizationally classified as 'Social health sciences' and 'Economics and business' (column 1 and 3, Fig 3). Publications which were organizationally classified as 'Economics and business' are cognitively related to developmental issues, software design, and food-related topics (i.e. published in journals devoted to these subjects). For these cases, it might be hypothesized that the authors were not

entirely knowledgeable of the journals which are regarded as standards in the field, and thus chose to publish in the wrong venue.

## 5. Discussion

A great deal of attention has been directed towards the issue of publishing in predatory journals. Studies have been conducted on selections of articles published in journals listed on blacklists, or sets of authors listed on these publications. Comparatively little is known, however, about the extent to which it poses a threat to scholars working at universities in western European countries or regions. Our analysis of publications submitted for inclusion in a national bibliographic database for SSH scholarship sheds additional light on the following understudied aspects: (i) How the number of publications in POA journals over time (2003–2016) relates to the total number of peer reviewed publications, with a focus on articles published in journals indexed by DOAJ, (ii) the disciplinary orientation of the journals, and (iii) authorship characteristics in terms of the number of authors listed, the experience level of the authors, the position of authors in the byline, and the organizational affiliation of the authors listed vs. the cognitive orientation of the journals published in.

Jeffrey Beall famously stated that POA was 'just one of the consequences of gold open access' [23]. While one could argue that the economic model of open access publishing has provided some sort of blueprint for a business model for 'predatory' open access publishers, we observe a discontinuity in yearly number of publications in POA journals. While the number of open access publications (as indexed by DOAJ) in VABB-SHW is on the rise, the number of POA publications has started to decline from 2012 onwards. This is an interesting finding as it suggests that scholars are increasingly aware of the issue and able to discern questionable outlets from 'genuine', qualitative open access journals. Awareness raising around the issue of POA has been carried out by the Flemish universities independently. They advise researchers to consult journal blacklists or whitelists, and use tools like ThinkCheckSubmit (https://thinkchecksubmit.org/). In addition, from 2014 onwards ECOOM has started to report publicly on the screenings for publications in questionable journals.

Our results further show that POA is not only an issue in developing countries. Publications in POA journals are submitted for inclusion in the VABB-SHW until the last year included for analysis. With regard to the fields to which the journals pertain, it is found that a high share is active in medicine related fields. For the social sciences, a disproportionately large number of POA journals pertains to the field of business and economics. Only one journal relates to a humanities field (philosophy). Differences in publishing behavior between disciplines could be a possible explanation for this. Engels et al. [45] have shown that publication patterns in the humanities differ from those in the social sciences (with specific attention for the case of VABB-SHW), with lower shares of journal article publications and higher shares of publications in Dutch and languages other than English.

Regarding predatory publishers active in medical and pharmaceutical fields, investigative journalists have found that such journals as well as their conferences are popular outlets for major pharmaceutical companies [46, 47]. When such institutions (knowingly or unknowingly) legitimize the business practices of POA journals, it becomes difficult for researchers to see through the façade of such shady practices. The fact that POA publishers have an additional, relatively large and financially powerful audience in, for example, the pharmaceutical field makes it all the more attractive for them to target these fields.

A study by Nicolas et al. [36] suggests that junior researchers show a more positive attitude towards open access and, as a consequence, they could thus be more at risk of publishing in POA journals. The authors contributing to POA publications analyzed here are junior

researchers as well as more experienced staff. Our results show that the contributions of senior researchers (measured as the average proportion of seniors listed in the byline) are in balance across the three sets of publications (POA, DOAJ indexed, and other peer-reviewed). We observe that the average proportion of seniors listed in the bylines of DOAJ indexed publications is somewhat lower than those for 'other peer-reviewed publications', and that the proportions for POA publications are somewhat in between these two sets. The assumption that these publications are authored mainly by inexperienced authors is therefore highly doubtful. Both junior and senior researchers should be targeted when raising awareness around the issue of POA.

We study author placement in the byline as another indicator for the importance of authors' contributions to a manuscript. For publications in POA journals with more than two authors, we find that the largest share has a junior author listed first. Knowing that all publications have a considerable share of senior researchers listed in the byline as well, we would like to emphasize the importance of mentoring. When it comes to selecting a venue where to publish one's research, seniors working together with junior scholars should direct their colleagues towards sources which, for example, list reliable journals or suggest tools like ThinkCheckSubmit. The fact that we observe a declining share of publications in questionable journals might relate to awareness raising in general (i.e. broad media attention, university awareness raising campaigns), as well as targeted awareness raising, e.g. informing authors when they have published in an outlet that is flagged as questionable, bringing up the issue of predatory publishing with new PhD students (i.e. during introductory courses in research ethics offered by doctoral education institutions), and editorials of disciplinary journals devoted to the problem of predatory publishing (see for example [47]). Scientific and professional associations could also develop guidelines and/or advertise existing tools to assist their members with avoiding questionable publishers (see for example the guidelines developed by the American Psychological Association, https://www.apa.org/monitor/2016/04/predatory-publishers).

## 6. Limitations and future research

The main motivation of this article was to demonstrate the screenings for POA conducted in light of the Flemish PRFS. While compiling the data and reviewing the literature, it was found that some important and yet understudied aspects could be analyzed with the data at hand. Although we study a relatively small set of publications, which is in a sense good news, important nuances were made with regard to disciplinary orientation of POA journals found and the authorship characteristics of the articles. For the latter we have limited ourselves to those authors present in the database. Two pragmatic considerations have guided our decision. First, it is not always easy to track authors unambiguously, and this is certainly the case for publications in POA journals. Second, there are no guarantees that the metadata listed on the websites of the POA journals or publishers are actually accurate. In some cases, for example, the journal's website, metadata or records of the articles were not available anymore.

Thus, only authors working at–or affiliated to a SSH research unit of one of the Flemish universities at the time of publication are included in our analysis. An important limitation relates to the latter; it is expected that we underestimate the number of publications in POA journals in other fields than the SSH. In depth analyses of, among other fields, engineering and technology and medical sciences would add more nuance to the findings presented here.

We have provided a quantitative analysis of the publications identified as POA. It would be interesting to contact the authors and address questions regarding their motivation for publishing in these journals (that is, if they were aware of the journal's quality), their knowledge of the reputable journals in the fields in which they've published, etc. Answers to these questions–as posed by one of the reviewers of this manuscript—might shed light on the reasons

why senior authors are also visibly present in our results. Are they less concerned with the journals they publish in? We leave this for future research, but wish to point out that some studies have already lifted a tip of the veil [48–49]. In addition, it would be interesting to further differentiate between career stages. Taking career milestones into account would deliver more nuance to the results presented in the above. This could be done by sending out surveys or screening curricula vitae. To provide more context for arguments made regarding the socio-economic divide between regions, comparative quantitative studies of other national or regional bibliographic databases in other countries would also provide interesting avenues for future research.

## 7. Conclusion

In summary, we have provided insight in the workings of the Flemish PRFS and how the yearly screenings for POA publications 'fit in'. We argue that such screenings are necessary, but that the usage of black- and whitelists should go hand in hand with expert assessment of journals and their publishers. There are large differences between individual journals (and publishers), but to be able to communicate and raise awareness around these issues, lists of journals and publishers come in handy. Our current analysis further indicates that PRFSs that include publication channels that are not included in international citation databases like WoS and Scopus (e.g. to allow better SSH coverage) need to pay particular attention to the presence of POA publications in these databases. Both locally maintained and international (citation) databases need to actively screen for POA journals and/or publishers to avoid legitimizing these practices. Openly communicating about the results of these screenings becomes an essential task for science administrators, policy advisors and professional organization alike.

## Supporting information

**S1 Fig. Distribution of POA journals (absolute counts) classified by Fields of Science coding scheme, level 2 (N = 144).**
(TIF)

## Acknowledgments

Disclaimer: This investigation has been made possible by the financial support of the Flemish government to the Centre for R&D Monitoring (ECOOM). The opinions in the paper are the authors' and not necessarily those of the government.

We would like to thank the two anonymous reviewers of this manuscript for their valuable suggestions. We thank Linda Sīle, for input on the study design and thoughts on an earlier version of the manuscript, which helped to improve the paper.

## Author Contributions

**Conceptualization:** Joshua Eykens, Raf Guns, A. I. M. Jakaria Rahman.

**Data curation:** Joshua Eykens, Raf Guns, A. I. M. Jakaria Rahman.

**Formal analysis:** Joshua Eykens, Raf Guns.

**Investigation:** Joshua Eykens, Raf Guns, A. I. M. Jakaria Rahman.

**Methodology:** Joshua Eykens, Raf Guns, Tim C. E. Engels.

**Project administration:** Joshua Eykens, Raf Guns.

**Resources:** Joshua Eykens, Raf Guns.

**Software:** Joshua Eykens, Raf Guns.

**Supervision:** Joshua Eykens, Raf Guns, Tim C. E. Engels.

**Validation:** Joshua Eykens, Raf Guns, Tim C. E. Engels.

**Visualization:** Joshua Eykens, Raf Guns.

**Writing – original draft:** Joshua Eykens, Raf Guns, A. I. M. Jakaria Rahman, Tim C. E. Engels.

**Writing – review & editing:** Joshua Eykens, Raf Guns, A. I. M. Jakaria Rahman, Tim C. E. Engels.

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
