## [Decision Letter · Decision Letter 0]

17 Jul 2019

PONE-D-19-14700

Identifying publications in questionable journals in the context of performance-based research funding

PLOS ONE

Dear Mr. Eykens,

Thank you for submitting your manuscript to PLOS ONE. After careful consideration, we feel that it has merit but does not fully meet PLOS ONE’s publication criteria as it currently stands. Therefore, we invite you to submit a revised version of the manuscript that addresses the points raised during the review process.

Two experts in the field have commented your paper, but they suggest to perform an exhaustive revision on it before considering its acceptance. Thus, my advice to the authors is to respond to each one of their queries and carry out all the improvements suggested by them. Personally, I believe this is a good paper, and might help scientists to improve their practices and standards in scientific publication. Nevertheless, an improved version of the manuscript should be revised for a second time by me and our reviewers before passing to a further step.

We would appreciate receiving your revised manuscript by Aug 31 2019 11:59PM. To enhance the reproducibility of your results, we recommend that if applicable you deposit your laboratory protocols in protocols.io, where a protocol can be assigned its own identifier (DOI) such that it can be cited independently in the future. For instructions see: http://journals.plos.org/plosone/s/submission-guidelines#loc-laboratory-protocols

We look forward to receiving your revised manuscript.

Kind regards,

Sergio A. Useche, Ph.D.

Academic Editor

PLOS ONE

Journal Requirements:

2) Please ensure that you refer to Figure 4 in your text as, if accepted, production will need this reference to link the reader to the figure.

3) Pleas provide a caption for Figure 4 in your manuscript.

4) Please include captions for your Supporting Information files at the end of your manuscript, and update any in-text citations to match accordingly. Please see our Supporting Information guidelines for more information: http://journals.plos.org/plosone/s/supporting-information.

5) We noted in your submission details that a portion of your manuscript may have been presented or published elsewhere. Please clarify whether this conference proceeding was peer-reviewed and formally published. If this work was previously peer-reviewed and published, in the cover letter please provide the reason that this work does not constitute dual publication and should be included in the current manuscript.

Reviewers' comments:

Reviewer's Responses to Questions

**Comments to the Author**

1. Is the manuscript technically sound, and do the data support the conclusions?

Reviewer #1: Yes

Reviewer #2: Yes

2. Has the statistical analysis been performed appropriately and rigorously? 

Reviewer #1: Yes

Reviewer #2: Yes

3. Have the authors made all data underlying the findings in their manuscript fully available?

Reviewer #1: Yes

Reviewer #2: Yes

4. Is the manuscript presented in an intelligible fashion and written in standard English?

Reviewer #1: Yes

Reviewer #2: Yes

5. Review Comments to the Author

Reviewer #1: This is an important topic to examine for public trust in scientific research. More specifically, a critical point the authors make is: whether initiatives to raise awareness about POA publishing have the adverse effect of ranishing all open access journals with the same brush."

I was left wondering why more 'predatory' publishers are active in the field of biomedical research and why POA is a serious problem in the neurosciences and related field. Tell me more about what is going on here. This is a worrisome finding.

How does prestige come into play with respect to the findings that "the proportions of senior authors in POA publications is in 8 out of 13 of the publication years higher than in other peer-reviewed publications"? Do senior authors need prestigious journals less?

The authors do a good job of following up on the research questions by addressing the implications of the answers to the research questions in the conclusion. Specifically, the emphasis on mentoring, and tools like ThinkCheckSubmit. In the U.S., university libraries are playing an educational (and non-judgemental role) in this space. Perhaps another recommendation could be, as part of dossier prep, that outlets are checked or verified. Certainly the conclusion that international citation databases screen for POA journals is an excellent step. I think reaching out to professional associations, such as AEA, COSSA, AMA, etc would also be helpful.

The suggestions for future studies, namely contacting authors to address questions about motivation and providing a comparative quantitative investigation of national and regional bibliographic databases, are valuable as well.

Reviewer #2: This is an important paper, as many have called for more action on the part of research funders with respect to this issue. Overall, this is a well-written paper. In particular, section 2.2 is really quite fascinating and relevant, as it excellently demonstrates some of the problems with relying on “white” and “black” lists, and the inconsistencies across the lists. My comments are primarily on areas that I feel could either use additional clarification or areas that I think would benefit from additional discussion.

Areas for additional discussion:

• In the introduction: why did you opt to focus on the SSH fields?

• In the discussion, regarding the generalizability of the data: Do you feel that your findings are reflective of other fields, particularly the medical sciences (which as you note, seems to be more affected by “POAs”?) What about other countries/geographic regions? In general, your findings regarding the decrease in researchers publishing in POA journals seems to be at odds with findings in other fields/locations. Why do you think this is?

• Following up on the last point, do you think that the decision by the GP to begin doing analyses of publications in POA journals by Flemish authors could be part of the reason for the drop-off in more recent years? (i.e. authors who may have been knowingly publishing in POA journals to boost their publication history lose some of their incentive because these publications will not be included in the VABB-SHW?)

• You note the huge discrepancy between using Beall’s list versus the CJB. Why do you think this is? Is the CJB not as comprehensive a list as Beall’s, or was it that Beall’s had several journals/publishers on it that were not actually POA?

• With regard to your findings on authorship (senior versus junior authors), I note that for junio researchers you have targeted those who are primarily pre-doctoral or early post-doctoral – presumably many of these researchers are still under the supervision of an experienced research mentor who may be helping them decide what journal(s) to publish in. Do you think this plays a role? Additionally, do you think you might see any differences if you further separated the “senior” researchers into the group of researchers that are late stage post-doctorates or have just recently been appointed to their first professor position (and looking to establish themselves and win grants), versus more experienced tenured professors that are possibly not under as much pressure to publish?

Areas for additional clarity:

• In the abstract, I suggest clarifying that this analysis was specific so the SSH fields.

• The introduction is quite comprehensive and provides good information, however, the literature review is quite lengthy and, in some places, it could be more concise. At times, the wordiness detracts from clarity.

• In Table 1, row 1, it appears that there are more journals on the blacklist than publications in the journals? (62 vs 59). I am also unclear whether the whitelist journals were used by ECOOM (and if so, how did they end of classifying journals that appeared on both white and black lists) or was the whitelist portion of the analysis was performed only by the authors of this paper?

• I would appreciate more clarity regarding how the list of POA journals was generated since, as the authors have noted, this is far from straight-forward. For example, since the list included journals over several years, what happens to a journal that is on a blacklist one year, but was removed from the list in a later year (or vice versa)? Or a journal that was on Beall’s list but not the CJB? What was the role of the whitelists in making this list? What if a journal was on both a black and a white list?

• In Table 2, and throughout the Results, I found it confusing what the difference is between the DOAJ-indexed publications and journals versus the “Peer-reviewed” publications and journals. Are peer-reviewed the total number of submissions to the VABB-SHW?

• I find the sentence beginning on line 409 and ending on 412 to require several read-throughs to be able to process. Is there a simpler way to say this?

• I found the heat map (Figure 4) to be a bit confusing. I assume one direction (i.e. column or row) corresponds to the field in which the authors work and one direction refers to the field in which the journal is classified, but its not clear which is which. I suggest better labeling on the figure and possibly re-phrasing the text. Furthermore, were there no journals that did not fit into a single classification (i.e. journals that accept all or multiple fields of research, etc.)?

• The line that begins on 496 – is this for POA journal articles or for all articles?

Finally, the article contains quite a few statements that would be stronger and/or more helpful if citations were provided to support them or for those who might want to seek further information. Examples of statements that would benefit from citations are those starting on the following lines:

• Line 47

• Line 48 (I also suggest elaborating on this as not all readers will be familiar with what this is referring to).

• Line 53 (what sources?)

• Line 64 (suggest providing citations for papers that document some of the unethical practices of “POAs”)

• Lines 225 – 230

• Line 485

6. PLOS authors have the option to publish the peer review history of their article (what does this mean?). If published, this will include your full peer review and any attached files.

Reviewer #1: No

Reviewer #2: No

---

## [Author Response · Author response to Decision Letter 0]

2 Sep 2019

Subject: Revision and resubmission of manuscript PONE-D-19-14700, Identitying publications in questionable journals in the context of performance-based research funding, PLoS ONE

Dear Dr. Useche,

Dear reviewers,

We are pleased to read that our manuscript will have merit for PLoS ONE. The suggestions offered by the reviewers have been very helpful; we appreciate the constructive and critical comments. We have included the reviewer comments in this letter, together with a description of how we addressed each concern. For each comment, we indicate where changes have been made. The revised manuscript as well as the marked-up copy with tracked changes were submitted via the editorial manger platform. 

Although most of the revisions prompted by the reviewers were addressed by making (often substantial) alterations to the text, we would like to further highlight an addition made to Table 1. In this table we present the results of the different screenings that were carried out over the past period. In the first row, indicating the screening for articles published between 2003 and 2012. For the numbers presented there, one of the reviewers has correctly noticed that the number of journals found on the blacklist was higher than the number of publications which appeared in the journals and were found in VABB-SHW. We have added a brief footnote to the figure which also refers readers to the corresponding report, but we here detail how this difference came about. A screening round is typically done on the basis of a journal list which is compiled from the raw bibliographical data we receive from the five Flemish universities. However, after compiling this list several operations happen that may cause differences between the list and the corresponding final version of the VABB-SHW (e.g. record consolidation, validation, corrections of publication year of some other variable, etc.). This may lead to such – at first sight at least – inaccuracies. 

A question raised by one of the reviewers regarding ‘predatory’ publishers active in the field of (bio)medical research and neurosciences has also been addressed in the text, but we wish to share some additional thoughts which might be out of scope of the manuscript but of interest for the reviewer. It is indeed worrisome finding that these fields suffer more from questionable publishers. This might have to do with, we believe, three things. First, the relative size and popularity of these fields. Second, differences in publishing practices between fields, and third, more potential interest of organizations and businesses in such fields to publish their findings regardless of their scientific value. 

With regard to the first aspect, it is reasonable to assume that predatory publishers are only interested in making ‘easy money’. The most efficient way to accomplish this, is to look for the largest market potential. Given the relative size of fields like biomedicine and the rapid growth of the neurosciences, these fields are interesting areas to operate in. This might be one of the reasons why prevalence is higher. Whereas we only briefly touched upon this in the literature overview, other authors have already shown that predatory publishers are targeting these fields intensively.

The second aspect closely relates to the former. These fields are known to differ in their publishing practices compared to, let’s say fields in the social sciences and humanities. Whereas SSH scholars are prone to publish, for example, more books, scholars from medicine related fields almost exclusively publish journal articles. In addition, within the medical fields, there is a greater prevalence of open access publishing in general. This is especially true for the preferred format of predatory publishers: the Gold Open Access model. 

Third, and this is again (much like the previous two aspects) speculative but has been shown by investigative journalists; employees of pharmaceutical and other companies have been publishing in predatory journals and sponsor predatory publisher’s conferences. Beall and colleagues also suggested that some pharmaceutical companies have an interest in publishing their findings, regardless of their scientific value. They do not really care where their research gets published, as long as it is out there and shows to potential new ‘costumers’ that their products are ‘beneficial’. 

As noted in the manuscript, this article elaborates considerably on an earlier conference proceeding. In the conference paper, we report on the yearly results of the screenings for POA journals and shortly address the architecture of the Flemish PRFS. It was presented as a research in progress paper and presented at the Science and Technology Indicators conference in Leiden, September 2018 (STI2018). The article has undergone external peer review and has been published in the conference proceedings. The proceedings book can be found here: http://sti2018.cwts.nl/proceedings and the paper can be found here: https://openaccess.leidenuniv.nl/handle/1887/65285. In the current paper however, we expand this description considerably. Table 1 of the current manuscript is copied from the conference proceedings, as it gives an overview of the results from our previous five screenings. In addition, we present a lot of new information, including novel bibliographic analyses, which haven’t been published or presented elsewhere. 

The other comments will be addressed below. We hope the revised manuscript will better suit PLoS ONE’s publishing standards. We would like to thank you for your continued interest in this study.

Sincerely, 

Joshua Eykens

ECOOM – University of Antwerp

Middelheimlaan 1

2020 Antwerp, Belgium

 

Reviewer comments, author responses and manuscript change

Reviewer 1

Comment 1

I was left wondering why more 'predatory' publishers are active in the field of biomedical research and why POA is a serious problem in the neurosciences and related field. Tell me more about what is going on here. This is a worrisome finding.

Answer: Besides the short note presented in the guiding letter above, in the discussion we have added some findings from a study by investigative journalists. 

See lines 507 – 513

Comment 2

How does prestige come into play with respect to the findings that "the proportions of senior authors in POA publications is in 8 out of 13 of the publication years higher than in other peer-reviewed publications"? Do senior authors need prestigious journals less?

Answer: This is a very interesting question and something we haven’t looked into. It is a reasonable assumption, but should be studied by making use of qualitative methods. We have added this in the limitations section and refer interested readers to studies which present more qualitative insights into the motivations of scholars who have published in questionable journals.

See lines 562 - 565

Comment 3

Perhaps another recommendation could be, as part of dossier prep, that outlets are checked or verified. Certainly the conclusion that international citation databases screen for POA journals is an excellent step. I think reaching out to professional associations, such as AEA, COSSA, AMA, etc would also be helpful.

Answer: Thank you for bringing this up, we didn’t think of including such suggestions but have done so now. This is indeed an important action which could certainly lead to greater awareness.

See lines 531 - 541

 

Reviewer 2

Comment 1

In the introduction: why did you opt to focus on the SSH fields?

Answer: This relates to the database at hand. VABB-SHW has been set up to better account for research in the SSH and as such offers a unique opportunity to study these fields in a comprehensive manner. We have made this more explicit in the abstract.

See line 25

Comment 2

In the discussion, regarding the generalizability of the data: Do you feel that your findings are reflective of other fields, particularly the medical sciences (which as you note, seems to be more affected by “POAs”?) What about other countries/geographic regions? In general, your findings regarding the decrease in researchers publishing in POA journals seems to be at odds with findings in other fields/locations. Why do you think this is?

Following up on the last point, do you think that the decision by the GP to begin doing analyses of publications in POA journals by Flemish authors could be part of the reason for the drop-off in more recent years? (i.e. authors who may have been knowingly publishing in POA journals to boost their publication history lose some of their incentive because these publications will not be included in the VABB-SHW?)

Answer: We belief the decrease in the number of POA publications has to do with (1) awareness raising in general as well as targeted awareness raising by Higher Education institutions. We have added some lines on these aspects. And (2) Cabells’ blacklisting service is relatively young and under continuous development. Publishers like Frontiers, which were included on Bealls list, are not present on Cabells’ Blacklist. This leads to quite large differences. 

In sum, we do think the screenings for publications in POA outlets have indeed increased awareness and may have lowered the prevalence. With regard to the possibility of authors who are knowingly using these journals to boost their CV’s; we have not found any alarming numbers of POA articles authored by a single individual. In the manuscript, we cannot disclose any information on the individual level, as this could be at odds with researchers’ privacy. 

 See lines 187, 374 - 376

Comment 3

You note the huge discrepancy between using Beall’s list versus the CJB. Why do you think this is? Is the CJB not as comprehensive a list as Beall’s, or was it that Beall’s had several journals/publishers on it that were not actually POA?

Answer: As noted above, this might have to do with the relative ‘newness’ of Cabells’ black list. Publishers like Frontiers, for example, are not present on Cabells black list, but many of their journals were listed on Bealls lists. We have added this example to illustrate this. 

 See line 187

Comment 4

With regard to your findings on authorship (senior versus junior authors), I note that for junior researchers you have targeted those who are primarily pre-doctoral or early post-doctoral – presumably many of these researchers are still under the supervision of an experienced research mentor who may be helping them decide what journal(s) to publish in. Do you think this plays a role? Additionally, do you think you might see any differences if you further separated the “senior” researchers into the group of researchers that are late stage post-doctorates or have just recently been appointed to their first professor position (and looking to establish themselves and win grants), versus more experienced tenured professors that are possibly not under as much pressure to publish?

Answer: With regard to mentoring; we have emphasized the importance of mentoring and other initiatives targeted at junior researchers. This has been done in the discussion. Regarding further differentiating between academic positions of authors; unfortunately we do not have any detailed information available on these aspects. We have added a brief note about this in the ‘Limitations and future research section’. 

See lines 531 – 541, 560 – 566 

Comment 5

In the abstract, I suggest clarifying that this analysis was specific so the SSH fields.

Answer: Thanks for pointing this out. We have made this more explicit in the abstract.

See line 25

Comment 6

The introduction is quite comprehensive and provides good information, however, the literature review is quite lengthy and, in some places, it could be more concise. At times, the wordiness detracts from clarity.

Answer: We have substantially altered both the introduction and the literature overview to increase readability. 

See section 1 and 2.3

Comment 7

In Table 1, row 1, it appears that there are more journals on the blacklist than publications in the journals? (62 vs 59). I am also unclear whether the whitelist journals were used by ECOOM (and if so, how did they end of classifying journals that appeared on both white and black lists) or was the whitelist portion of the analysis was performed only by the authors of this paper?

Answer: Regarding the discrepancy in table 1; besides the note presented in the guiding letter, we have added a footnote to the table further detailing how this difference came about. With regard to the whitelist used by ECOOM; we have further detailed how this list is being curated in the methods section. 

See lines 132 – 135, 271 - 283

Comment 8

I would appreciate more clarity regarding how the list of POA journals was generated since, as the authors have noted, this is far from straight-forward. For example, since the list included journals over several years, what happens to a journal that is on a blacklist one year, but was removed from the list in a later year (or vice versa)? Or a journal that was on Beall’s list but not the CJB? What was the role of the whitelists in making this list? What if a journal was on both a black and a white list?

Answer: Thank you for pointing out this rather confusing aspect. We have included some more details on this procedure in two places in the methods section.

See lines 271 – 283, 295 – 300

Comment 9

In Table 2, and throughout the Results, I found it confusing what the difference is between the DOAJ-indexed publications and journals versus the “Peer-reviewed” publications and journals. Are peer-reviewed the total number of submissions to the VABB-SHW?

Answer: We have further detailed this in the methods section. 

See lines 295 - 300

Comment 10

I find the sentence beginning on line 409 and ending on 412 to require several read-throughs to be able to process. Is there a simpler way to say this?

Answer: This sentence has been rephrased.

See lines 430 - 435

Comment 11

I found the heat map (Figure 4) to be a bit confusing. I assume one direction (i.e. column or row) corresponds to the field in which the authors work and one direction refers to the field in which the journal is classified, but its not clear which is which. I suggest better labeling on the figure and possibly re-phrasing the text. Furthermore, were there no journals that did not fit into a single classification (i.e. journals that accept all or multiple fields of research, etc.)?

Answer: Indeed, some journals did not fit into a single category. These are indicated by the label ‘General/very broad’ (last row). The heatmap has been relabeled and the caption has been rephrased.

 See figure 3, lines 463 - 464

Comment 12

The line that begins on 496 – is this for POA journal articles or for all articles?

Answer: Thanks for pointing this out, this has been clarified in the text. 

See lines 526 - 527

Comment 13

Finally, the article contains quite a few statements that would be stronger and/or more helpful if citations were provided to support them or for those who might want to seek further information. Examples of statements that would benefit from citations are those starting on the following lines:

Comment 13.1

Line 47

Answer: This sentence has been omitted, since after all it does not add any value to the introduction. 

Comment 13.2

Line 48 (I also suggest elaborating on this as not all readers will be familiar with what this is referring to).

Answer: We have added some information about the report to the introduction. 

See lines 50 – 54

Comment 13.3

Line 53 (what sources?)

Answer: Here we were referring to statements by public opinion makers in Flanders, shared primarily on social media. After reconsideration, these do not add context or value to the manuscript, so we have omitted these lines.

Comment 13.4

Line 64 (suggest providing citations for papers that document some of the unethical practices of “POAs”)

Answer: A citation has been added here.

See line 65

Comment 13.5

Lines 225 – 230

Answer: Reference has been added. 

See line 226

Comment 13.6

Line 485

Answer: This sentence has been altered and a reference has been added.

See lines 507

---

## [Decision Letter · Decision Letter 1]

17 Oct 2019

Identifying publications in questionable journals in the context of performance-based research funding

PONE-D-19-14700R1

Dear Dr. Eykens,

We are pleased to inform you that your manuscript has been judged scientifically suitable for publication and will be formally accepted for publication once it complies with all outstanding technical requirements.

With kind regards,

Sergio A. Useche, Ph.D.

Academic Editor

PLOS ONE

Additional Editor Comments (optional):

Reviewers' comments:

Reviewer's Responses to Questions

**Comments to the Author**

1. If the authors have adequately addressed your comments raised in a previous round of review and you feel that this manuscript is now acceptable for publication, you may indicate that here to bypass the “Comments to the Author” section, enter your conflict of interest statement in the “Confidential to Editor” section, and submit your "Accept" recommendation.

Reviewer #1: All comments have been addressed

Reviewer #2: All comments have been addressed

2. Is the manuscript technically sound, and do the data support the conclusions?

Reviewer #1: Yes

Reviewer #2: Yes

3. Has the statistical analysis been performed appropriately and rigorously? 

Reviewer #1: Yes

Reviewer #2: Yes

4. Have the authors made all data underlying the findings in their manuscript fully available?

Reviewer #1: Yes

Reviewer #2: Yes

5. Is the manuscript presented in an intelligible fashion and written in standard English?

Reviewer #1: Yes

Reviewer #2: Yes

6. Review Comments to the Author

Reviewer #1: The author has adequately address my concerns and those of the other reviewer. The manuscript provides solid work on an important question.

Reviewer #2: (No Response)

7. PLOS authors have the option to publish the peer review history of their article (what does this mean?). If published, this will include your full peer review and any attached files.

Reviewer #1: No

Reviewer #2: No

---

## [Editor Report · Acceptance letter]

31 Oct 2019

PONE-D-19-14700R1 

Identifying publications in questionable journals in the context of performance-based research funding 

Dear Dr. Eykens:

I am pleased to inform you that your manuscript has been deemed suitable for publication in PLOS ONE. Congratulations! Your manuscript is now with our production department. 

With kind regards,

on behalf of

Dr. Sergio A. Useche 

Academic Editor

PLOS ONE